# Leadership, Mental Toughness, and Attachment Relationship in the World Beach Volleyball Context

**Nayara Malheiros Caruzzo** [1,*] **, João Ricardo Nickenig Vissoci** [2]**, Andressa Ribeiro Contreira** [3]**, Aryelle Malheiros Caruzzo** [4] **and Lenamar Fiorese** [1]

[1] Department de Educação Física, Universidade Estadual de Maringá, Maringá 87020-900, Brazil; lenamarfiorese@gmail.com
[2] Duke Global Health Institute (DGHI), Duke University, Durham, NC 27708, USA; joaovissoci@gmail.com
[3] Escola Superior de Ciências da Saúde, Universidade do Estado do Amazonas, Manaus 69065-001, Brazil; acontreira@uea.edu.br
[4] Centro de Treinamento Paralímpico Brasileiro, São Paulo 04329-000, Brazil; aryelle_malheiros@hotmail.com
* Correspondence: nayaramalheiros@gmail.com

**Abstract:** For a long time, competitive sport has focused only on aspects related to performance. However, studies in social psychology have indicated the importance of focusing on the human development of athletes, which can occur through training environments that promote psychological well-being. Thus, this study aimed to analyze the impact of the coach-athlete attachment style, mediated by the coach's leadership style, on the mental toughness of athletes in the world beach volleyball context. Elite beach volleyball athletes (*n* = 65), participants of the World Tour 2018, were part of the study. The Coach-Athlete Attachment Scale (CAAS), Mental Toughness Index (MTI) and Leadership Scale for Sport (LSS) were used as instruments. For data analyses we used polychoric correlation and a bias-corrected factor score path analysis. Path analysis showed that perceived secure attachment was positively associated with athletes' mental toughness (0.24; 0.31; 0.25), but leadership styles did not mediate this relationship. For athletes with anxious attachment profiles, the perception of autocratic leadership style was associated with athletes' mental toughness (1.01; *p* = 0.03), when their interaction style is focused on coaching-instruction. It concludes that the secure attachment relationship can bring increases in levels of athletic mental toughness, whereas for athletes with insecure attachment, the autocratic style was shown to be associated with the highest levels of mental toughness.

**Keywords:** sport; beach volleyball; athlete; attachment; mental toughness

## 1. Introduction

The search for resources that enhance performance under challenging conditions, trying to reduce the negative side effects of the elite sport system, is part of the athletic routine. To maintain a high level of performance under a wide variety of stressors is the goal of every high-performance athlete. Among the development of human psychological resources that enable athletes to use their cognitive, emotional, and behavioral mechanisms to prevent psychological problems, aiming to maintain or improve their performance is mental toughness [1,2]. An important construct for athletes in sports such as beach volleyball, who face adversity in matches, in which, as a rule, substitutions, which could serve as strategies to improve the team's performance, are not allowed.

The most well-known sports model consists of prioritizing performance and success above the physical and mental health aspects of athletes. However, in the light of the aspects presented, research developed with Olympic medalist athletes of this modality pointed out that the most effective techniques to overcome stress and adversities were cognitive ones [3,4], reinforcing the relevance of the psychological resources, as the mental toughness. Specifically, it is an intentional, flexible, and efficient subsidy that aims at

the enactment and maintenance of the proposed goals [5], gathering personal resources that help the individual to achieve consistent performance in the presence of different contextual demands [6].

Its conceptualization has been discussed in the sense of understanding mental toughness as a stable personality trait or as being a mindset that can vary between situations and over time [6,7]. A systematic review pointed out that toughness can come 50% from personal characteristics, while the environment can act strongly on its development in the same way [7].

In this sense, the promotion of psychological skills to deal with the demands of the sport context, specifically, is considered a function of sport psychologists. However, it is observed that sports coaches play an important role in the psychological and emotional development of athletes through their daily interactions in the training environment and competitions [8–10]. Its role in the socio-emotional construction of the athlete finds support in Attachment Theory, which provides a psychological framework that contributes significantly to the understanding of emotional bonds formed in close relationships, as well as those that permeate the sport context [11].

Evidence points out that the sports coach can be an attachment figure, similar to the role played by parents and teachers in childhood, being seen by young people as "the strongest and wisest" ([12], p. 1457). Through the individual's emotional connections to the attachment figure, three different attachment styles can be established: secure, avoidant, and anxious attachment [13]. Secure attachment is considered a positive emotional attachment, presented by individuals who have quality connections with the attachment figure, are more psychologically adjusted, have lower levels of stress, and are characterized by experiencing positive emotions such as relief, joy, and gratitude. In contrast, individuals with attachment styles considered insecure, displayed in the form of anxious and avoidant styles, tend to present negative working models, feeling unworthy of love, indifferent to the attachment figure, repeating the interactions they had during their childhood, in which they experienced inconsistent behaviors with their caregivers (anxious style) or even rejection during times of need (avoidant style) [14].

It is also noteworthy that attachment styles can help explain the formative processes of athletes [15] since they shape the subjects' mental representations of themselves and others, implying a mapping of the subject since attachment experiences shape people's self-image, who rely on attachment experiences as a source of information to learn about themselves [16–18]. Besides the individual attachment styles, which allow the individual to predict their future relationships, the way the coach establishes a relationship with the athlete is also associated with their leadership profile, since it shapes their behavior profile through the decision-making and interaction styles they use with their team [19].

According to the Multidimensional Model of Leadership [20], leadership is seen within a relational interaction process, since it is proposed that the leader considers the situational characteristics, the leader's characteristics, and those of the members that make up the group. The team's performance and satisfaction are considered, the result of the leader working harmoniously in relation to these three aspects of leader behavior (required, actual, and preferred). Thus, coaches work both through interpersonal relationships and instrumental orientations, which direct the goals and provide structure to their teams [19].

Regarding leader attachment profiles, a study conducted with military personnel and their followers identified important aspects [21]. The results showed that the more avoidant the leaders' profiles were, the more they were described as less skilled at dealing effectively with the military's emotion, and the more anxious, the less skilled at congratulating the group on accomplishing tasks. Anxious leaders showed a negative effect on soldiers' instrumental functioning, mediated by a lack of effectiveness in dealing effectively with task-focused situations, indicating that anxious attachment strongly interferes with successful completion of group tasks, which in turn erodes the confidence of the led, as an anxious leader's doubts about his or her own ability echoes over the athletes' abilities to complete tasks.

In the face of the presented considerations, the relevance of investigations about the investigated variables and their implications on the athletes' capacity to face the demands of the performance sport context can be observed. However, despite the arguments and scientific evidence previously presented, no studies were found hypothesizing the association between the coach-athlete attachment relationship, coach leadership, and athletic mental toughness, and this is a gap that the present study intends to fill. Such an investigation is scientifically grounded by considering that coaches have the potential to become an attachment figure in the sport environment [11], that the way coaches interact with their athletes are related to the attachment style adopted between the figures (coach and athlete), and that positive environments have implications on the strategies and mechanisms that influence the process of mental toughness development [22].

Thus, the present study aims to contribute to filling the gap presented by analyzing the impact of the coach-athlete attachment style, mediated by the coach's leadership style, on the mental toughness of athletes in the context of world beach volleyball. As conceptual hypotheses, we have that: (a) secure attachment perceptions, mediated by democratic leadership style and social support interaction, lead to increased mental toughness and that (b) insecure attachment perceptions (anxious and avoidant), when mediated by an autocratic style and coach-instruction-oriented interaction, may impact athletic mental toughness. Such considerations are supported in the literature, which has evidenced positive relationships between autonomy-promoting behaviors and positive perceptions about the coach-athlete relationship and negative trajectories between controlling coach behaviors and constructs of the coach-athlete relationship [8,23].

## 2. Materials and Methods

### 2.1. Participants

The study included elite beach volleyball athletes ($n$ = 65) participating in the 2018 World Tour, held in China, the United States, and Mexico, the last of which was a female-only event. The stages are classified as four-star (China and the United States) and three-star (Mexico) competitions, considered one of the main competitions of the sport worldwide. The event includes delegations from all over the world and the participation of up to five pairs per country, in each gender (male and female).

Each stage has the participation of 64 pairs (128 athletes) of each sex, totaling 256 athletes. However, we adopted as inclusion criteria athletes who were proficient in English, and it was not possible to include Chinese, Japanese, Thai, Russian, and Czech athletes due to language limitations. Therefore, all athletes who met the inclusion criteria ($n$ = 180) were invited to participate in the study. Some refused to participate voluntarily in the research ($n$ = 65) and others did not return the questionnaires to the researcher ($n$ = 45). The incorrect completion of the questionnaire items was an exclusion criterion ($n$ = 5), totaling the sample in 65 athletes (38 women and 27 men), representing Australia, Austria, Brazil, Canada, Chile, Colombia, Spain, United States, France, Greece, Italy, Latvia, Lithuania, Mexico, Norway, Poland, Qatar, Czech Republic, Russia, Serbia, Slovakia and Switzerland, with an average age of 28.16 years (dp = 4.85).

### 2.2. Instruments

#### 2.2.1. Coach-Athlete Attachment Scale (CAAS)

The coach-athlete attachment style was assessed through the Coach-Athlete Attachment Scale (CAAS), in its English version [24]. The questionnaire assesses the attachment relationship between athlete and coach, and was used in its original version. The scale is composed of 19 items, distributed in three dimensions: Avoidant Style (e.g., "I do not turn to my coach/athlete for reassurance"); Anxious Style (e.g., "Sometimes I worry that my athlete/coach is not committed to me, as I am to him/her") and Secure Style (e.g., "I know I can count on my coach/athlete"). Responses are given on a 7-point Likert scale, ranging from "Strongly Disagree" (1) to "Strongly Agree" (7). The score of each dimension is calculated by means of the arithmetic mean of the items that compose it, and for its

interpretation, it is considered that the highest values in a given dimension represent the athlete's predominant attachment style.

### 2.2.2. Mental Toughness Questionnaire

The Mental Toughness Index (MTI) scale [6] was used in its original version (English) with the athletes. The instrument is unidimensional, composed of eight items that assess how athletes think, feel, and behave in their sport, in this case, beach volleyball. The MTI contains items such as "I believe in my ability to achieve my goals" and "I constantly overcome adversity." Answers are given on a seven-point Likert scale, ranging on a continuum from "False 100% of the time" (1) to "True 100% of the time" (7). The score is calculated from the average of the sum of its component items. Higher values mean greater mental strength.

### 2.2.3. Leadership Scale for Sport (LSS)

To assess the coaches' leadership style, the Leadership Scale for Sport (LSS) [20] was used. The scale that assesses the athletes' perception of their coach's leadership style was used, and was applied in the English version. The scale is composed of 40 questions, distributed in five dimensions. Two are based on the coach's decision-making style: "Autocratic" and "Democratic", and three are based on the coach's interaction style: "Social support", "Reinforcement", and "Coaching-instruction". Responses are given on a 5-point Likert-type scale, ranging from "Never" (1) to "Always" (5). The score is calculated from the average sum of the items that make up each dimension.

### 2.3. Collection Procedures

After the approval by the Standing Committee on Ethics in Research with Human Beings (SCETHB) regarding the ethical and methodological aspects of scientific research with human beings (Opinion No. 1,324,411/2015), the collection was carried out in the second half of 2018, by the researcher herself. The athletes participating in the stages of the World Tour, held respectively in China, the United States, and Mexico, were invited to participate in the research. Those who fit the inclusion criteria were given the questionnaires, along with the Informed Consent Form. The collection was carried out at the competition venues, with an average duration of 15 min for filling out the scales.

### 2.4. Data Analysis

Data analysis was performed using the R studio Statistical Language Program, version 3.3. The descriptive results were presented as mean (M) and standard deviation (SD) and absolute and relative frequency. We performed a polychoric correlation, represented graphically by a network using the qgraph and igraph packages in R Studio.

Finally, a trajectory analysis similar to structural equation analysis was applied, using a sequential mediation model, in order to verify to what extent the relationship with the coach and their leadership style can contribute to mental robustness. The bias-corrected factor score path analysis (BCFSPA) is a robust, yet little-known alternative for estimating Structural Equation Modeling (SEMs). BCFSPA is a maximum likelihood estimator that addresses several important limitations of full information methods in SEMs, while improving conventional methods with limited data and information [25].

The choice for such a model is justified since the coefficient estimates generated from the BCFSPA analysis were practically unbiased in each instance and uniformly showed more consistency compared to the results presented by the SEM model, which is supported by the literature [25]. The conventional estimator uses the factor score variance-covariance matrix to estimate structural relationships, while the corrected approach, introduces measurement model-based adjustments to correct the factor score variance-covariance matrix and thus provide an unbiased estimate of the true variance-covariance matrix. For its implementation, four steps, described below, are required.

First, to implement the BCFSPA estimator, we separately estimated the confirmatory factorial models for each latent variable (attachment, decision style and leadership interaction, and mental toughness)-step a. Next, we needed to structure the connections between the latent variables to estimate the coefficients between the trajectories. To do this, we first predicted the factor scores for each latent using the regression method. Subsequently, a variance and covariance matrix was estimated with the generated factorial scores (step b).

Next, we corrected the covariance matrix and standardized solutions with the scale of each latent variable were considered, thus defining the variance of the variable for the measurement model. To implement these corrections with the factor scores obtained through the regressions, the covariance between each set of scores was divided by the respective products of the factor scores and factor loadings (step c). Finally, a typical path analysis was implemented to obtain coefficient estimates for the structural model. Specifically, a path analysis was conducted with the corrected covariance matrix by entering a simulated sample covariance matrix ($n = 100$), correcting the sample size equal to the number of cases in our data set (step d). We chose $n = 100$ since Kelcey identified it as a sample N of higher reliability and robustness after several simulations with different sample numbers [25].

For this, a conceptual model, based on Attachment Theory [11] and the Multidimensional Model of Leadership (adapted from Chelladurai and Saleh [20]) was hypothesized (Table 1), in which the Attachment Style, Decision Style and Interaction Style correspond to the latent variables, which were combined, according to the theoretical hypotheses, into nine different models (three for each attachment style).

**Table 1.** Conceptual hypothetical structural equation of sequential mediation with trajectories of each model. Source: the authors.

| Structural Equation Models | Attachment STYLE | Leadership Decision Style | Leadership Interaction Style | Mental Toughness |
|---|---|---|---|---|
| Model 1 | Secure | Democratic | Social Support | Mental Toughness |
| Model 2 | Secure | Democratic | Coaching-instruction | Mental Toughness |
| Model 3 | Secure | Democratic | Positive Reinforcement | Mental Toughness |
| Model 4 | Anxious | Autocratic | Social Support | Mental Toughness |
| Model 5 | Anxious | Autocratic | Coaching-instruction | Mental Toughness |
| Model 6 | Anxious | Autocratic | Positive Reinforcement | Mental Toughness |
| Model 7 | Avoidant | Autocratic | Social Support | Mental Toughness |
| Model 8 | Avoidant | Autocratic | Coaching-instruction | Mental Toughness |
| Model 9 | Avoidant | Autocratic | Positive Reinforcement | Mental Toughness |

The illustrative model shows four latent variables: attachment style, coach's decision-making style, coach's leadership interaction style, and mental toughness (Figure 1). We chose to use sequential mediation, which allows us to detail the extent to which the influence of attachment (secure, anxious, or avoidant) on mental toughness operates through the leadership style (autocratic and democratic) and the interaction style offered by the coach (social support, coaching-instruction, and positive reinforcement). Statistically, the sequential mediation process was summarized as the product of the relationships between attachment and leadership style (path labeled as a in Figure 1), leadership style and interaction (path labeled as b in Figure 1), and interaction and mental toughness (path labeled as f in Figure 1). Thus, Model 1 aims to test the impact of secure attachment on mental toughness mediated by the democratic style and social support offered by the coach. Models 2 and 3 use "Coaching-instruction" and "Positive reinforcement" as the leadership interaction style, respectively, following the same sequential mediation order presented in Model 1.

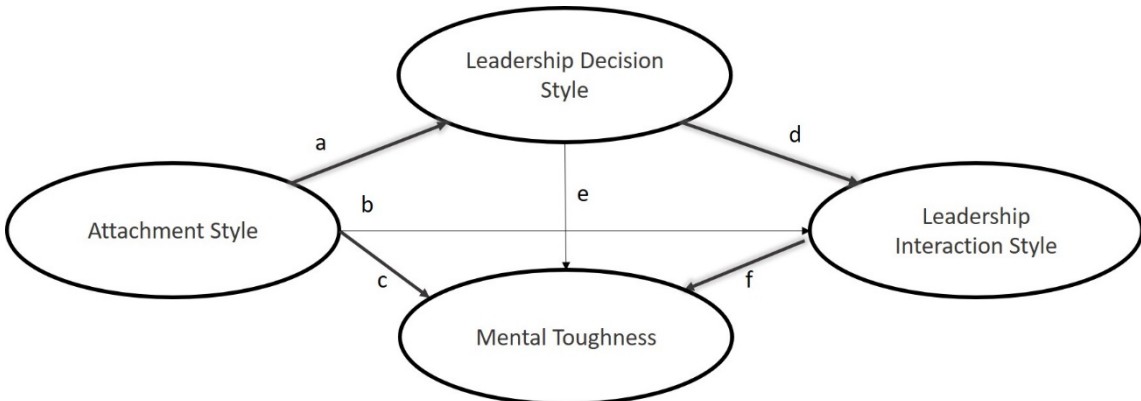

**Figure 1.** Conceptual hypothetical structural equation model of sequential mediation with labeled trajectories.

Models investigating the impact of insecure attachment were applied in Models 4 to 9. Anxious attachment was tested in Models 4, 5, and 6, using as trajectories: anxious attachment -> autocratic style -> social support (Model 4)/coaching-instruction (Model 5)/positive reinforcement (Model 6) -> mental toughness (Table 1).

Finally, avoidant attachment was tested in Models 7, 8, and 9, following the same order as the models presented previously: avoidant attachment -> autocratic style -> social support (Model 7)/coaching-instruction (Model 8)/positive reinforcement (Model 9) -> mental toughness (Table 1).

## 3. Results

Table 2 below presents the descriptive results of the sample that made up this study ($n$ = 65). It can be noticed that most (58.5%) of the subjects are female, with an average time of experience of 9.57 (5.63) years and with previous sports experience prior to a career in beach volleyball (93.8%).

**Table 2.** Characterization of world beach volleyball athletes ($n$ = 65).

| Sociodemographic Variables | |
| --- | --- |
| Age (years), Mean (dp) | 28.16 (4.85) |
| Female, *n* (%) | 38 (58.5) |
| Time of Experience (years), Average (dp) | 9.57 (5.63) |
| Time with partner team (years), Average (dp) | 2.89 (3.39) |
| Time in indoor volleyball, Average (dp) | 8.5 (4.24) |
| Previous sports experience, *n* (%) | |
| Yes | 61 (93.8) |
| No | 4 (6.2) |
| Outcome Variables | |
| Attachment, Average (dp) | |
| Anxious | 2.28 (1.45) |
| Avoidant | 3.32 (1.33) |
| Secure | 5.69 (1.39) |
| Leadership, Average (dp) | |
| Coaching-instruction | 2.09 (1.01) |
| Social Support | 2.63 (0.97) |
| Positive Reinforcement | 2.01 (1.1) |
| Autocratic | 3.62 (1.02) |
| Democratic | 2.41 (0.91) |
| Mental Toughness, Average (dp) | 5.74 (0.62) |

Regarding the investigated variables, the attachment with the highest prevalence perceived by the athletes was secure attachment (M = 5.69), followed by avoidant attachment (M = 3.32). The leadership decision style adopted by coaches most commonly noted by

athletes was autocratic (M = 3.62), while the most commonly perceived decision style was social support (M = 2.63). Mental toughness was perceived as relatively high (M = 5.74), considering that the maximum mean value in the scale score is 7.00 points.

In Figure 2, we observe the network of associations between the sociodemographic variables, attachment styles, athlete mental toughness, and athlete-perceived coach leadership style. The nodes represent the variables, while the edges indicate the positive (represented by the green edges) and negative (red edges) associations. The proximity between the nodes and the thickness of the edges represent the strength of the associations.

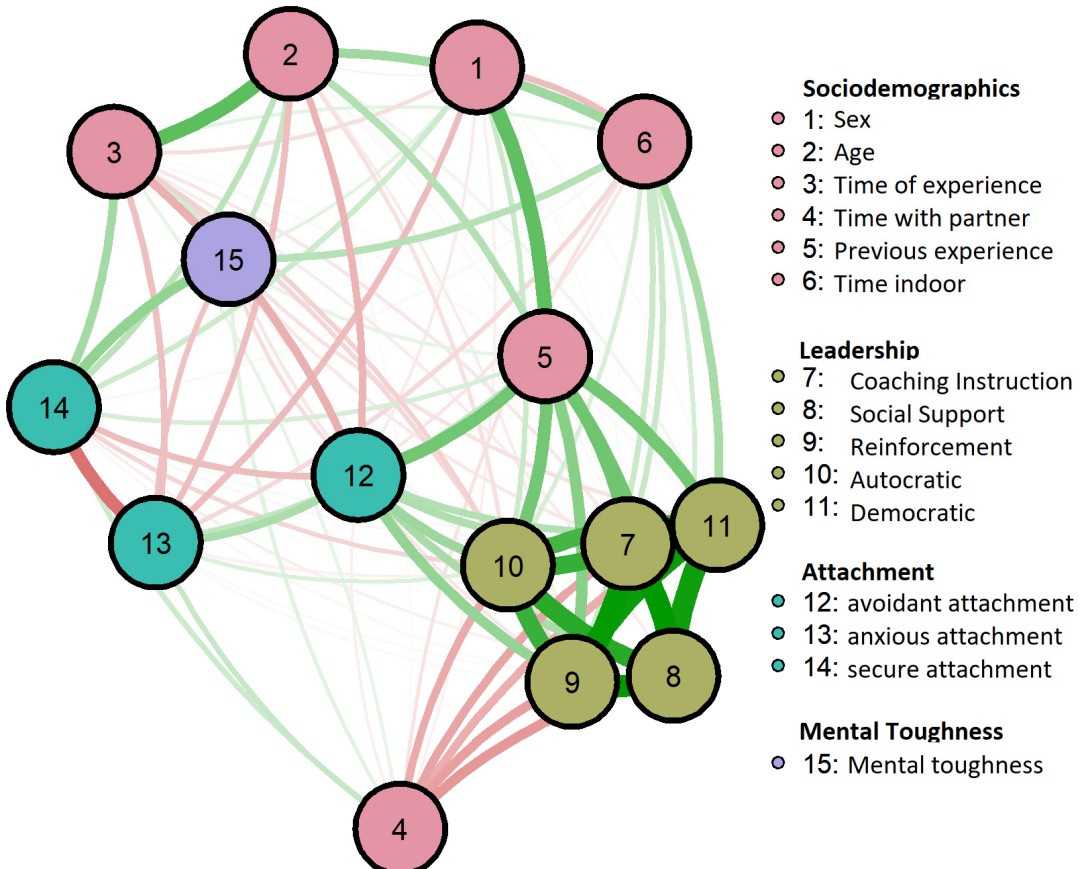

**Figure 2.** Representation of the associations between sociodemographic variables, coach-athlete attachment relationship, perceived leadership styles, and athletic mental toughness, by means of a network.

The mediation effect and each path, generated from the BCFSPA estimators, are presented in Table 3, indicating the impact of secure attachment on mental toughness, mediated by the democratic leadership style and the leadership interaction styles, as indicated in Models 1, 2, and 3.

The democratic decision-making style associated positively with social support interaction style (0.85), coaching-instruction (0.91), and positive reinforcement (0.91) significantly ($p < 0.001$) (Table 3). Perceived secure attachment associated positively with athletes' mental toughness (0.24; 0.31 and 0.25, respectively; Table 2), but leadership styles did not mediate the relationship between perceived attachment and mental toughness.

Next, in Table 4, we present the impact of anxious attachment on mental toughness, mediated by autocratic leadership style and the respective interaction styles, indicated according to Model 4, 5, and 6.

**Table 3.** Results of the structural model on the impact of secure attachment on the mental toughness of beach volleyball athletes, mediated by the democratic leadership style and the interaction styles social support (Model 1), coaching-instruction (Model 2) and positive reinforcement (Model 3), based on the BCFSPA analysis.

| Structural Model | Model 1 Social Support | | | Model 2 Coaching-Instruction | | | Model 3 Positive Reinforcement | | |
|---|---|---|---|---|---|---|---|---|---|
| **Trajectories** | **Estimate** | **SE (B)** | **_p_-Value** | **Estimate** | **SE (B)** | **_p_-Value** | **Estimate** | **SE (B)** | **_p_-Value** |
| a | −0.12 | 0.09 | 0.21 | −0.15 | 0.10 | 0.12 | −0.12 | 0.10 | 0.21 |
| b | 0.02 | 0.05 | 0.64 | −0.02 | 0.04 | 0.64 | −0.02 | 0.04 | 0.68 |
| c | 0.24 | 0.09 | 0.01 * | 0.31 | 0.10 | 0.001 * | 0.25 | 0.09 | 0.01 * |
| d | 0.85 | 0.05 | <0.001 * | 0.91 | 0.04 | <0.001 * | 0.91 | 0.04 | <0.001 * |
| e | −0.21 | 0.18 | 0.24 | 0.06 | 0.23 | 0.79 | 0.03 | 0.24 | 0.90 |
| f | 0.25 | 0.18 | 0.16 | −0.04 | 0.23 | 0.85 | −0.03 | 0.24 | 0.91 |
| Mediation effect: a-d-f | −0.03 | - | - | 0.01 | - | - | 0.01 | | |

Note: SE (B) are standard errors. BCFSPA is the corrected path analysis from the factor analysis. The trajectory a = Attachment-Decision style, b = Attachment-Interaction style, c = Attachment-Mental toughness, d = Decision style-Interaction style, e = Decision style-Mental toughness, f = Interaction style-Mental toughness. * Significant values for $p < 0.05$. Source: the authors.

**Table 4.** Results of the structural model on the impact of anxious attachment on mental toughness of beach volleyball athletes, as mediated by autocratic leadership style and the interaction styles social support (Model 4), coaching-instruction (Model 5), and positive reinforcement (Model 6), based on BCFSPA analysis.

| Structural Model | Model 4 Social Support | | | Model 5 Coaching-Instruction | | | Model 6 Positive Reinforcement | | |
|---|---|---|---|---|---|---|---|---|---|
| **Trajectories** | **Estimate** | **SE (B)** | **_p_-Value** | **Estimate** | **SE (B)** | **_p_-Value** | **Estimate** | **SE (B)** | **_p_-Value** |
| A | 0.11 | 0.09 | 0.24 | 0.11 | 0.09 | 0.24 | 0.20 | 0.09 | 0.04 * |
| B | −0.04 | 0.03 | 0.21 | −0.09 | 0.02 | <0.001 * | 0.03 | 0.02 | 0.08 |
| C | −0.06 | 0.09 | 0.52 | −0.18 | 0.11 | 0.09 | −0.10 | 0.10 | 0.31 |
| D | 0.96 | 0.03 | <0.001 * | 0.98 | 0.02 | <0.001 * | 0.98 | 0.02 | <0.001 * |
| E | −0.59 | 0.34 | 0.09 | 1.01 | 0.45 | 0.03 * | 1.00 | 0.57 | 0.08 |
| F | 0.62 | 0.34 | 0.09 | −1.02 | 0.45 | 0.02 * | −0.99 | 0.57 | 0.08 |
| Mediation effect: a-d-f | 0.07 | - | - | −0.16 | | | −0.19 | | |

Note: SE (B) are standard errors. BCFSPA is the corrected path analysis from the factor analysis. The trajectory a = Attachment-Decision style, b = Attachment-Interaction style, c = Attachment-Mental toughness, d = Decision style-Interaction style, e = Decision style-Mental toughness, f = Interaction style-Mental toughness. * Significant values for $p < 0.05$. Source: the author.

The autocratic leadership style of the coach was associated with social support (0.96), coaching-instruction (0.98), and positive reinforcement (0.98), indicating that the decision-making style adopted by the coach does not suggest a specific form of interaction with the athletes. The perception of autocratic leadership style was associated with athletes' mental toughness (1.01; $p = 0.03$) when their interaction style is geared towards coaching-instruction, indicating that, for athletes with an anxious attachment profile, the coach's decision style seems to be more strongly linked to increased mental toughness than to the interaction style. This is because direct perception of coaching-instruction-oriented interaction was negatively associated with mental toughness (−1.02).

Next, in Table 5, we present the impact of avoidant attachment on mental toughness, mediated by autocratic leadership style and the respective interaction styles, indicated according to Models 7, 8, and 9.

Perceiving the attachment style as avoidant decreases athletes' perceived social support and coaching-instruction-oriented interactions (trajectory b = −0.13 and −0.14, respectively; Table 4) and positively associates with adopting an autocratic leadership style, regardless of the interaction style adopted (trajectory a = 0.50, 0.50, and 0.59; <0.001; Table 5). Adopting autocratic leadership increases the interaction of social support, coaching-

instruction, and positive reinforcement (trajectory d, Table 4). Finally, like anxious attachment (see Table 3), subjects with perceived avoidant attachment perceive increases in their mental robustness (1.46) when coaches adopt an autocratic leadership style-amidst the coaching-instruction interaction. However, the perception of the coaching-instruction leadership style adopted by the coach, which advocates behavior geared toward improving athlete performance by focusing on hard and demanding training in the techniques and tactics of the sport, decreases the mental toughness of the athlete with perceived avoidant attachment by 1.36 (Table 5).

**Table 5.** Results of the structural model on the impact of avoidant attachment on mental toughness of beach volleyball athletes, mediated by autocratic leadership style and the interaction styles social support (Model 7), coaching-instruction (Model 8), and positive reinforcement (Model 9), based on BCFSPA analysis.

| Structural Model | Model 7Social Support | | | Model 8Coaching-Instruction | | | Model 9Positive Reinforcement | | |
|---|---|---|---|---|---|---|---|---|---|
| **Trajectories** | **Estimate** | **SE (B)** | ***p*-Value** | **Estimate** | **SE (B)** | ***p*-Value** | **Estimate** | **SE (B)** | ***p*-Value** |
| a | 0.50 | 0.09 | <0.001 * | 0.50 | 0.09 | <0.001 * | 0.59 | 0.08 | <0.001 * |
| b | −0.13 | 0.03 | <0.001 * | −0.14 | 0.02 | <0.001 * | 0.01 | 0.02 | 0.48 |
| c | −0.04 | 0.12 | 0.73 | −0.32 | 0.13 | 0.01 * | −0.15 | 0.12 | 0.22 |
| d | 1.02 | 0.03 | <0.001 * | 1.04 | 0.02 | <0.001 * | 0.98 | 0.02 | <0.001 * |
| e | −0.55 | 0.40 | 0.17 | 1.49 | 0.52 | 0.00 * | 1.12 | 0.57 | 0.05 |
| f | 0.60 | 0.37 | 0.11 | −1,36 | 0.49 | 0.00 * | −1,05 | 0.57 | 0.06 |
| Mediation effect: a-d-f | 0.31 | - | - | −0.71 | - | - | −0.60 | - | - |

Note: SE (B) are standard errors. BCFSPA is the corrected path analysis from the factor analysis. The trajectory a = Attachment-Decision style, b = Attachment-Interaction style, c = Attachment-Mental toughness, d = Decision style-Interaction style, e = Decision style-Mental toughness, f = Interaction style-Mental toughness. * Significant values for $p < 0.05$. Source: the author.

## 4. Discussion

Our literature searches verified that this is the first study to investigate the impact of the coach-athlete attachment relationship mediated by the coach's leadership styles on the mental toughness of beach volleyball athletes in the global context. The results partially confirmed the hypothesis of the study, as we observed that secure attachment style can lead to an increase in mental toughness (0.30) of athletes in this context, but in a way that is not mediated by the coach's leadership style (Table 2). Similarly, the hypothesis that this impact on athletes' mental toughness would be mediated by the coach's democratic leadership style was refuted, as the autocratic style indicated a 1-point increase in mental toughness when in the midst of the coaching-instruction interaction and anxious attachment relationship (Tables 4 and 5).

Both athletes with perceptions of anxious and avoidant attachment relationships who perceive their coaches as autocratic showed a positive association with mental toughness, partially confirming the second conceptual hypothesis of this study, since this association occurred in a direct way and was not mediated by the coach-instruction interaction style adopted by the coach. The literature shows that the individual's current relationship pattern is the result of a very consistent pattern of interactions with attachment figures during childhood, and that the most representative interaction models become part of the implicit knowledge and tend to operate automatically and unconsciously [26]. In this sense, it is inferred that the relationship pattern for these athletes works in a similar way to the relationship they had with their parents in childhood: a decision mode that requires little participation from children, still seen by caregivers as incapable of such attitudes. Thus, they remain immersed in relationships seen as desirable, but with interactions that are not very accessible to the figures, a peculiar characteristic to the development of insecure attachment [16].

In contrast, for these same individuals with insecure attachment, the perception of the interaction focused on training-instruction seemed to contribute negatively to mental

toughness, thus refuting the conceptual hypothesis of the study. This receives support in attachment theory, as it shows that the doubts of individuals with anxious attachment about their own ability reflect on the instrumental abilities of those involved in the relationship [21]. Subjects with a perception of avoidant attachment may have difficulties in accepting the support received, which aims, through coaching-instruction, to coordinate and structure the activities of athletes during athletic sessions and routines, since it goes against the idea of superiority and autonomy of these individuals [27].

More broadly, leaders with insecure attachment focus on their own insecurities and dissatisfactions, and may not respond to the needs of their subordinates. They draw too much attention to themselves, failing to promote perception of competence in their followers, causing them to doubt their own capabilities [26]. It is inferred that, perhaps because of this, the perception of an interaction focused on instrumental activities, such as coaching-instruction, has been negatively associated with mental toughness in athletes with an anxious and avoidant attachment profile in this study.

Still, knowing that mental toughness is, in part, adaptable and develops over time may suggest that coaches are influential in the process of developing this construct through interventions in the context in which the subjects are involved, such as the training environment [6]. It is pointed out that a positive coach-athlete relationship and constructive coach behavior can support athletes to strengthen their ability to overcome challenges [28]. Such findings support the results of the present study, which indicate the association of only the secure attachment style with mental robustness, with no such associations identified with anxious and avoidant attachment. This happens because this influence on the athletes can happen in a more powerful way, when effective bonds are created between the figures involved, since the coach can act as a safe haven and source of comfort, which promotes security in times of need [11].

It was noted that the democratic and autocratic leadership styles were positively associated with the three leadership interaction styles (social support, coaching-instruction, and positive reinforcement), regardless of the perceived attachment style (trajectory d; Tables 3–5). This demonstrates that the form of decision-making adopted by the coach does not predict their form of interaction with the athletes. Various coach behaviors can be expected in order to fit different athletic situations [20]. In this sense, the Multidimensional Model of Leadership, proposed by Chelladurai [29], specifies that the effectiveness of the leader's behavior will result from the congruence between the members' preferences, the situational characteristics, and the particular characteristics of the leader.

This may be the explanation for the positive associations between the perceived leadership decision-making and interaction styles of the athletes. Regardless of the perceived attachment relationship and the autocratic or democratic style adopted, coaches seem to make use of this congruence, which aligns with leadership expectations, as a result of the situation, the members' preference, and their personal characteristics [30]. Thus, even though controlling behaviors can be associated with demotivation [31] and have negative effects on the coach-athlete relationship and how atheletes respond to adversity [8], positive outcomes are more likely when the coach is able to align all three leadership behaviors [30], as this can lessen the effects of behaviors that do not provide support during training [8].

Although the present study provides empirical evidence on the importance of leadership aspects as mediators of the relationships between attachment and mental robustness in world beach volleyball athletes, some limitations need to be pointed out. Among the limitations is the use of the scale only in the English version, which limited the participation of athletes who were not proficient in the language. It is noteworthy that, despite this limitation, the use of the scales in a globally known language allows for the advancement of scientific knowledge about the attachment relationship between coach and athlete, its perceived leadership aspects, and the mental toughness construct, still little explored in the sport. Finally, although the data collection took place at three different moments (three stages of the world circuit), the cross-sectional study format allowed for significant predictions about the relationships between the variables, but did not allow for causal

relationships, whereas a longitudinal study would allow for more robust inferences. Finally, it is also suggested that future research investigating leadership aspects should investigate the preference style of the athletes since in the present study only perception was investigated.

## 5. Conclusions

This study revealed that a secure attachment relationship between coach and athlete can bring increases in the levels of athletic mental toughness, which partially confirms the conceptual hypothesis of the study, since this association occurred in a direct way and not mediated by the leadership styles of the coach. For athletes with insecure attachment, the autocratic style proved to be associated with higher levels of mental toughness, which also partially confirms the hypothesis of the study, since this result was observed in the direct trajectory between autocratic style and mental toughness.

It is noteworthy, however, that the perception of leadership interaction focused on instrumental activities, such as coaching-instruction, was perceived as negatively associated with mental toughness for athletes with anxious and avoidant attachments. Indicating that insecurity about their capabilities is reflected in anxious athletes and that, for avoidant athletes, demonstrating the need and accepting such help does not match their mental representation of autonomy and superiority.

The findings of this study contribute for coaches and others involved in the sport context to understand the importance of the emotional bonds formed in these environments for the development of psychological skills to cope with competitive and social demands. Such contributions extend to the use of various leadership interactions with athletes, which seek to promote sport environments that are positive to athletic development and help to find a balance between competitive demands and personal athletic goals. Specifically in the beach volleyball context, the figure of the coach becomes even more important, since teams are small (compared to conventional team sports) and participants on the world tour spend months away from home, and the coach might play a role as the only source of personal support.

**Author Contributions:** Conceptualization, N.M.C.; Vissoci and Fiorese; methodology, N.M.C. and J.R.N.V.; formal analysis, N.M.C.; investigation, A.R.C. and A.M.C.; data curation, A.M.C. and J.R.N.V.; Writing—Original draft preparation, N.M.C. and A.R.C.; Writing—Review and Editing, A.M.C.; visualization, J.R.N.V.; supervision, L.F.; project administration, N.M.C. and L.F.; funding acquisition, N.M.C. and L.F. All authors have read and agreed to the published version of the manuscript.

**Funding:** This study was financed in part by the Coordenação de Aperfeiçoamento de Pessoal de Nível Superior-Brasil (CAPES)-Finance Code 001.

**Institutional Review Board Statement:** The study was conducted according to the guidelines of the Declaration of Helsinki, and approved by the Ethics Committee of State University of Maringá (protocol code 1.324.411/2015).

**Informed Consent Statement:** Informed consent was obtained from all subjects involved in the study.

**Data Availability Statement:** Not applicable.

**Acknowledgments:** To all subjects for participating in the study and to a Volleyball Brazilian Confederation for support during the word championships where data collect occurred.

**Conflicts of Interest:** The authors declare no conflict of interest. The funders had no role in the design of the study; in the collection, analyses, or interpretation of data; in the writing of the manuscript, or in the decision to publish the results.

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
