# Peer review of "Leadership, Mental Toughness, and Attachment Relationship in the World Beach Volleyball Context"

_sustainability, doi:10.3390/su131910748_

Round 1

Reviewer 1 Report

The article is original. It brings new knowledge important for sports practice and also for the development of sports sciences. I positively evaluates the theoretical background and used literature. I evaluate the research goals and the methods used positively. The research results are processed clearly. The discussion is at the required level. Overall, I rate the article very positively.

Author Response

We appreciate the reviewer's considerations.

Reviewer 2 Report

1. The aim is clear; it is clear what the study found. 
2. The findings of this study are crucial and the references are relevant, 
3. It is clear what is already known about this topic, the research question is clearly outlined; the research question was justified given what is already known about the topic.
4. The process of subject selection is clear; the variables defined and measured are appropriate, the study methods are valid and reliable
5. The data is presented in an appropriate way, tables are relevant and clearly presented, titles, columns, and rows were labeled correctly and clearly.
6. I am clear about what is a statistically significant result.
7. The results are discussed from multiple angles and placed into context.
8. The conclusions answer the aims of the study.

Author Response

(The authors gave the same response as above.)

Author Response

(The authors gave the same response as above.)

Reviewer 4 Report

Content:

This is a study examining the influence of the relationship between the coach-athlete attachment and the coach's leadership style on the mental toughness of athletes.

The authors examined 65 elite beach volleyball athletes who participated in the World Tour 2018. To measure the coach-athlete attachment style, the coach's leadership style and mental toughness, the authors used the Coach-Athlete Attachment Scale, the Mental Toughness Index and the Leadership Scale for Sport. For data analyses, the authors calculated correlations and performed a bias-corrected factor score path analysis.

The results showed that perceived secure attachment was positively associated with mental toughness. For athletes with anxious attachment profiles, the perception of autocratic leadership style was associated with athletes' mental toughness. Therefore, the authors concluded that a secure attachment relationship can increase the levels of athletic mental toughness. However, in athletes with insecure attachment, the autocratic style was shown to be associated with the highest levels of mental toughness.

Comments:

The manuscript is interesting and well written. The major limitations which are the cross-sectional design and the fact that many athletes had to be excluded on the basis of insufficient proficiency in English.

My only concern are few misspellings which change the meaning of the manuscript. For example, in the title it says "in the word beach volleyball context." I guess the authors mean "world" instead of "word." I also think that "world beach volleyball" does not specify the fact that these were top athletes. Therefore, I ask myself whether it would be more precise to write "during a world beach volleyball championship." The authors should think about the title again. However, more importantly, they should check the whole manuscript for misspellings which lead to the confusion of words. This needs to be done by a researcher, not by a computer program, because misspellings which produce a correct word won't be picked up by a computer.

There are also few unnecessary hyphens within words. These need to be removed prior to a potential publication.

Author Response

Dear Editor, the requested changes have been highlighted in red in the text.
The manuscript underwent spelling revision, according to your guidelines.
Thanks for your contributions.

Reviewer 5 Report

Thank you for the opportunity to read this manuscript. The authors did extensive work to contribute to the field of social psychology. However, this work has a fundamental flaw which is an example of low quality scientific methodology. It is incomprehensible for me that authors collected answers/data via an English questionnaire for non-speaking English participants! I cite you "we adopted as inclusion criteria athletes who
were proficient in English, and it was not possible to include Chinese,
Japanese, Thai, Russian, and Czech athletes due to language limitations". How did authors check English proficiency, by asking them do they speak and write English?? Each validated questionnaire must be presented to each participant in their native language. All participants have four stages of the response process when answering a questionnaire 1) need to understand the question, 2) retrieve relevant information, 3) integrate the information to form a judgment, 4) map their response to the available response options. This is very difficult to process for non-native speakers of the survey language especially when questions are formulated to assess conative functions in humans. Therefore, these results may have low data quality. 

Author Response

Dear editor, General requests for correction were met and highlighted in red in the manuscript,
seeking to improve its presentation quality. We appreciate your suggestions.

Round 2

Reviewer 5 Report

No comments